# The impact of comorbid severe mental illness and common chronic physical health conditions on hospitalisation: A systematic review and meta-analysis

**Naomi Launders**[1]\*, **Kate Dotsikas**[1], **Louise Marston**[2], **Gabriele Price**[3], **David P. J. Osborn**[1,4], **Joseph F. Hayes**[1,4]

**1** Division of Psychiatry, UCL, London, United Kingdom, **2** Department of Primary Care and Population Health, UCL, London, United Kingdom, **3** Health Improvement Directorate, Public Health England, London, United Kingdom, **4** Camden and Islington NHS Foundation Trust, St Pancras Hospital, London, United Kingdom

\* naomi.launders.19@ucl.ac.uk

## Abstract

### Background

People with severe mental illness (SMI) are at higher risk of physical health conditions compared to the general population, however, the impact of specific underlying health conditions on the use of secondary care by people with SMI is unknown. We investigated hospital use in people managed in the community with SMI and five common physical long-term conditions: cardiovascular diseases, COPD, cancers, diabetes and liver disease.

### Methods

We performed a systematic review and meta-analysis (Prospero: CRD42020176251) using terms for SMI, physical health conditions and hospitalisation. We included observational studies in adults under the age of 75 with a diagnosis of SMI who were managed in the community and had one of the physical conditions of interest. The primary outcomes were hospital use for all causes, physical health causes and related to the physical condition under study. We performed random-effects meta-analyses, stratified by physical condition.

### Results

We identified 5,129 studies, of which 50 were included: focusing on diabetes (n = 21), cardiovascular disease (n = 19), COPD (n = 4), cancer (n = 3), liver disease (n = 1), and multiple physical health conditions (n = 2). The pooled odds ratio (pOR) of any hospital use in patients with diabetes and SMI was 1.28 (95%CI:1.15–1.44) compared to patients with diabetes alone and pooled hazard ratio was 1.19 (95%CI:1.08–1.31). The risk of 30-day readmissions was raised in patients with SMI and diabetes (pOR: 1.18, 95%CI:1.08–1.29), SMI and cardiovascular disease (pOR: 1.27, 95%CI:1.06–1.53) and SMI and COPD (pOR:1.18, 95%CI: 1.14–1.22) compared to patients with those conditions but no SMI.

**Data Availability Statement:** All relevant data are within the manuscript and its Supporting Information files.

**Funding:** This study was supported by Public Health England (PhD2019/002 - NL), the Wellcome Trust (211085/Z/18/Z - JFH), the Medical Research Council (MC\PC\17216 - DPJO), University College London Hospitals NIHR Biomedical Research Centre (NL, DPJO, JFH) and the NIHR ARC North Thames Academy (DPJO, JFH). This report is independent research supported by the National Institute for Health Research ARC North Thames. The funders had no role in study design, data collection and analysis, decision to publish, or preparation of the manuscript. The views expressed in this publication are those of the author(s) and not necessarily those of the National Institute for Health Research, Public Health England, or the Department of Health and Social Care

**Competing interests:** The authors have declared that no competing interests exist.

## Conclusion

People with SMI and five physical conditions are at higher risk of hospitalisation compared to people with that physical condition alone. Further research is warranted into the combined effects of SMI and physical conditions on longer-term hospital use to better target interventions aimed at reducing inappropriate hospital use and improving disease management and outcomes.

## Introduction

People with severe mental illness (SMI) have more physical health comorbidities [1–5] and poorer prognoses from those comorbidities [6] than the general population. Physical health comorbidities can lead to reduced quality of life [7], worsening mental health [8], and drives excess mortality in people with SMI [9, 10].

Previous systematic reviews have found that people with SMI are at a higher risk of 30-day readmissions compared to those without SMI [11, 12], and that those with SMI and physical health comorbidities are at higher risk of psychiatric admissions compared to those with SMI alone [13].

Studies based on hospital records alone have found that people with SMI use hospitals for physical health more frequently than people without SMI for emergency admissions [14], preventable admissions [15] and all-cause admissions [16]. However, without accounting for underlying physical comorbidities, whether this represents inappropriate use of services is unclear. A recent meta-analysis by Ronaldson et al. [17] found that in studies controlling for physical health comorbidities there were more hospitalisations, ED visits and longer length of stays in people with SMI compared to those without SMI, suggesting the higher service use is not explained by higher prevalence of physical health conditions alone.

The relationship between physical and mental health and the effect on service utilisation is likely complex, dependent on a range of patient and provider factors. Known drivers of hospital utilisation in the general population, such as poor medication adherence, polypharmacy [18] or inappropriate prescribing [19], continuity of care, and patient satisfaction [20–22] may influence hospital utilisation differently depending on the number and type of underlying mental and physical health conditions in a population.

In order to understand the effect of having both a diagnosis of SMI and of physical health conditions on hospital utilisation, we undertook a systematic review and meta-analysis of observational hospital utilisation studies, comparing people with SMI and one of five common physical long-term conditions (LTCs), compared to those with either SMI or LTCs alone. These diseases (cardiovascular diseases, chronic obstructive pulmonary disease (COPD), cancers, diabetes and liver disease) were chosen because of their high burden of disease globally and/or their impact on those with SMI.

## Methods

### Search strategy

We searched the following sources on 24 March 2020 for publications or grey literature within the remit of the study without date restrictions: PubMED, EmBase, Web of Science, PsychInfo, PsychExtra, Health Management Information Centre. Searches for new publications were performed on 17 December 2020 and 17 March 2022. Searches included terms for severe mental

illness, physical health conditions and hospitalisation (S1 Appendix). We performed forward and backward citation searching of relevant studies, reviews and editorials. Where conference abstracts were identified searches for related articles were performed. Conference abstracts were excluded from the final analysis, though those with available data were included in a sensitivity analysis. The study protocol was registered with Prospero: CRD42020176251.

## Outcomes

The primary outcomes were planned or unplanned hospital admissions, for either all-causes, all physical health causes, causes specific to the physical LTC under study, or ambulatory care sensitive conditions (ACSC), a list of conditions for which emergency admission is thought to be avoidable [23]. Secondary outcomes were readmissions and attendance at EDs or other acute outpatient care for these causes.

## Inclusion and exclusion criteria

We included observational studies of adults under the age of 75, managed in the community, and diagnosed with SMI and at least one of the physical LTCs of interest (cardiovascular diseases, COPD, cancers, diabetes and liver disease). We defined SMI as patients with a diagnosis of either schizophrenia, bipolar disorder or other non-organic long-term psychotic disorders, in line with the Quality Outcomes Framework used by the NHS in England [24]. We therefore excluded studies that included major depression in their definition of SMI, without stratifying results by mental health condition.

We excluded studies without comparator populations, interventional studies, and reviews. We also excluded studies focused solely on children and young people (under 18) or the elderly (over 75 years), or in populations not managed in the community. We excluded studies focused on planned outpatient care, preventative services such as cancer screening where the setting of service provision was unclear and context specific, and studies focused on admissions for specific procedures. Finally, we excluded studies where the outcome was hospitalisation for a specific physical health condition other than the physical LTC of interest.

## Data screening and extraction

We collated the results of the literature search using EndNote X9 (Clarivate Analytics, PA, USA) and removed duplicates. The first researcher (NL) screened titles and abstracts against inclusion and exclusion criteria in Microsoft Access, and records obtained in March 2020 (70%) were screened by the second researcher (KD). We resolved disagreements through discussion and calculated the Kappa statistic for inter-rater agreement. We acquired full text articles for all studies identified for inclusion which were screened by the first researcher and a 20% sample was screened by the second researcher. We extracted data from included studies using a standardised form, which was piloted on a sub-set of articles prior to finalisation. This form included variables describing the study focus (exposure, outcome, study population, location); design (methodology, effect measure and size, matching or adjusting variables, follow up time, study period), and publication (publication year).

## Statistical analysis

We analysed the data both as a narrative synthesis, and a meta-analysis stratified by physical LTCs. Studies providing adjusted odds ratios (OR) or hazard ratios (HR) were included in the meta-analyses. Pooled OR and HR were calculated on aggregate data and the relationship between SMI and physical health and secondary care utilisation quantified using a random

effects meta-analysis, performed in R [25] and R Studio [26]. In-study bias was be assessed using Newcastle-Ottawa scale (NOS) assessment for observational studies. We assessed publication bias by visual scrutiny of funnel plots of effect size against standard error, and where more than ten studies were considered, using an Egger's test. Study heterogeneity was measured using the $I^2$ statistic [27]. We undertook subgroup analysis to account for SMI diagnosis group and outcome measures. Where differences were found between groups in subgroup analysis, meta-regression was performed to determine the effect of controlling for these groups on heterogeneity. We performed a sensitivity analysis using three-level hierarchical meta-analysis. This method allows for the inclusion of multiple results from single studies, accounting for variance between participants and between studies as in random effects meta-analysis, but also the variance between multiple effect sizes within a study [28].

## Results

We identified 5,129 records, of which 3,646 remained after deduplication (Fig 1). Inter-rater agreement of title and abstract screening was 91.4%, with a Kappa statistic of 0.57. Following screening, 50 studies [29–78] were included in the narrative synthesis, published between 2006 and 2022 (Table 1).

### Study characteristics

Most studies were conducted in the United States (US) (n = 33; Table 1). Forty-four studies quantified the risk of admissions, readmissions or ED visits in a patient population (median population size: 53,343; interquartile range (IQR): 23,856–185,981); while in five studies the focus was the number of index admissions which resulted in a readmission (median admissions: 184,898, IQR: 132,604–581,469), and one investigated the admission ratio of 4,275 ED visits. The majority of studies (n = 38) included adults with an age range of 20 to 65 or wider, while seven focused on those over the age of 65. The remaining studies excluded patients under the age of 30 or 40 (n = 3), those over the age of 50 (n = 1) or those over 35 (n = 1). The included studies were heterogeneous in population, exposure, outcome, and effect measure and 27 could be stratified into multiple analyses based on these factors (Table 2). Of the 104 unique analyses, 59 investigated inpatient admissions over at least a year, with a median follow up of five years (IQR: 2–14). A further 27 investigated inpatient admissions limited to a 28 to 31 day period following an index admission (termed 30-day readmissions) and 12 investigated ED visits (median follow up: 2 years, IQR: 2–5 years). Two analyses investigated 7-day readmissions, two investigated 90-day readmissions, one combined inpatient admissions and ED visits over a ten year period, and one calculated the odds of admission in those attending an ED (Table 2). ED use was the only acute outpatient care outcome identified, and we did not identify any studies of planned inpatient admissions.

### Study quality and risk of bias

The majority of studies had pre-existing psychiatric illness as a focus (n = 37/50), while 11 considered a broad range of risk factors for hospital admission, of which SMI was one. Two studies included SMI as a covariate for a different exposure of interest. The majority (n = 42) of studies were in unmatched populations and 11 did not provide adjusted effect measures. Ten studies were limited to a single region of a country, and two to single hospitals (S1 Table). Denominator populations were sourced from hospital records in 31 studies, hospital and outpatient or pharmacy records in eleven and primary care records in eight (S1 Table).

Of the 39 studies which provided adjusted effect estimates, 37 controlled for age and gender, one controlled for gender but not age [38] and one controlled for age and was limited to

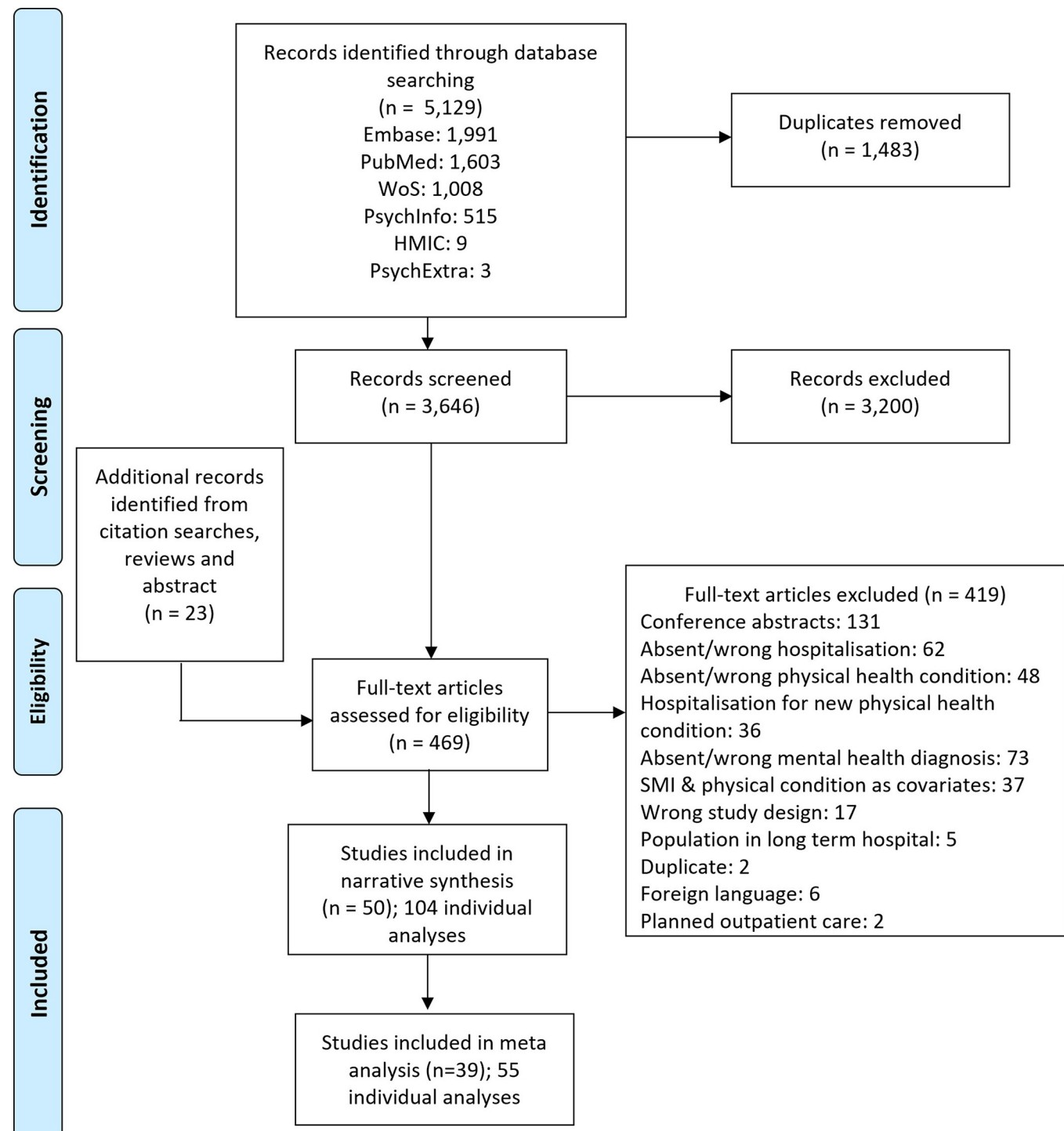

**Fig 1. PRISMA flow chart.** WoS: Web of Science; HMIC: Health management information consortium.

the female population only [64]. Thirty-three studies controlled for physical health comorbidities and eight for prior healthcare utilisation (S1 Table). Almost half the studies (n = 24/50) had a NOS of between 6 and 7 (fair quality), while 19 had a score of 8 or 9 (high quality) and seven had a score of under 6 (poor quality; S1 and S2 Tables). Two studies with multiple

**Table 1. Study description.**

| Authors | Pub year | Study design | Exposure | Outcome | Population | Notes | Study period | Follow up | Pop size | Unit of measure | Country | Area | Age | Matched |
|---|---|---|---|---|---|---|---|---|---|---|---|---|---|---|
| **Studies of diabetes and SMI** | | | | | | | | | | | | | | |
| Egglefield et al. [29] | 2020 | Cross sectional | Antipsychotic adherence | Preventable diabetes admissions | Medicaid registered patients with diabetes | Unadjusted data provided for patients with schizophrenia | 2012 | 1 year | 191,521 | Person | US | One region | 18–64 | No |
| Helmer et al. [30] | 2020 | Cohort | SMI and other MH conditions | Any, acute and chronic ACSC admissions | Veterans Affairs registered patients with diabetes | | 2010 | 1 year | 151,614 | Person | US | National | >66 | No |
| Stockbridge et al. [31] | 2019 | Cross sectional | Schizophrenia, bipolar disorder, and other MH conditions | Diabetes admissions | Insured patients with diabetes | | 2011–2013 | 3 years | 229,039 | Person | US | National | 20–64 | No |
| Tsai et al. [32] | 2019 | Cohort | Bipolar disorder | Hyperglycaemia admissions | Patients with diabetes | | 1999–2013 | Up to 11 years | 30,477 | Person | Taiwan | National | Adults | Yes |
| Goueslard et al. [33] | 2018 | Cohort | Schizophrenia | Acute diabetes complications long-term readmissions | Patients with type 1 diabetes | | 2009–2012 | 3 years | 45,655 | Person | France | National | 15–35 | No |
| Edwards et al. [34] | 2014 | Cohort | Home-Based Primary Care | ACSC admissions | Veterans Affairs registered patients with diabetes | Psychosis is a covariate | 2006–2010 | Up to 5 years | 56,608 | Person | US | National | >67 | No |
| Druss et al. [35] | 2012 | Cross sectional | Schizophrenia, bipolar disorder, and other MH conditions | ACSC admissions | Medicaid registered patients with diabetes | | 2003–2004 | 2 years | 657,628 | Person | US | National | < = 65 | No |
| Leung et al. [36] | 2011 | Cohort | Schizophrenia, bipolar disorder, and other MH conditions | Diabetes admissions | Medicaid or medicare registered patients with type 2 diabetes | | 2005 | 1 year | 106,174 | Person | US | One region | >18 | No |
| Mai et al. [37] | 2011 | Cohort | Schizophrenia, affective psychosis, other psychoses, and other MH conditions | Diabetes admissions | Patients with diabetes | | 1990–2006 | Up to 15.5 years | 43,671 | Person | Australia | One region | >18 | Yes |
| Cramer et al. [38] | 2010 | Cross sectional | Risk factors and comorbidities | More than one all-cause long-term readmission | Medicaid registered patients with diabetes | Psychosis is one of many risk factors considered | 2005 | 1 year | 695 | Person | US | National | Adults | No |
| Yan et al. [39] | 2019 | Cohort | Risk factors and comorbidities | All-cause admissions | Patients with antipsychotic-treated schizophrenia, bipolar 1 disorder or major depressive disorder | Type 2 diabetes is one of many risk factors considered | 2013–2016 | 1 year | 38,195 | Person | US | Multiple regions | >18 | No |
| Chen et al. [40] | 2012 | Cohort | Outpatient quality of care | All-cause 30-day readmissions | Commercially insured patients with diabetes | Psychosis is a covariate | 2010 | 30 days | 30,139 | Person | US | National | >19 | No |
| Guerrero Fernandez de Alba et al. [41] | 2020 | Cohort | Schizophrenia, and other MH conditions | All-cause and diabetes admissions and ED attendances | Patients with type 2 diabetes | | 2012 | 1 year | 63,365 | Person | Spain | One region | >18 | No |
| Chwastiak et al. [42] | 2014 | Cohort | SMI | All cause 30-day and long-term readmissions | Patients with diabetes | | 2010–2011 | 30 days / up to 2 years | 82,060 | Person | US | One region | >18 | No |
| Becker et al. [43] | 2011 | Cohort | Schizophrenia | Hyperglycaemia or hypoglycaemia admissions or ED attendances | Patients with diabetes | | 1996–2006 | 1–10 years | 5,033 | Person | Canada | One region | 18–50 | Yes |
| Krein et al. [44] | 2006 | Cross sectional | SMI | All-cause admissions | Veterans Affairs registered patients with diabetes | | 1997–1998 | 1 year | 36,546 | Person | US | National | Mean 58 | Yes |
| Kurdyak et al. [45] | 2017 | Cohort | Schizophrenia | Diabetes and all-cause admissions and ED attendances | Patients with diabetes | | 2011–2013 | 2 years | 1,131,375 | Person | Canada | One region | 19–105 | No |
| Shim et al. [46] | 2014 | Cohort | Schizophrenia or diabetes | Diabetes and all-cause ED attendances | Medicaid registered patients with diabetes and/or schizophrenia | | 2006–2007 | 2 years | 432,112 | Person | US | Multiple regions | 18–64 | No |
| Sullivan et al. [47] | 2006 | Cross sectional | Bipolar disorder, and other MH conditions | Admissions in those attending ED for diabetes | Patients with diabetes | | 1994–1998 | 4.5 years | 4,275 | Admissions | US | Single site | >18 | No |

*(Continued)*

**Table 1.** (Continued)

| Authors | Pub year | Study design | Exposure | Outcome | Population | Notes | Study period | Follow up | Pop size | Unit of measure | Country | Area | Age | Matched |
|---|---|---|---|---|---|---|---|---|---|---|---|---|---|---|
| Wang et al. [78] | 2021 | Cohort | SMI | All cause admissions | Patients with diabetes | | 2000–2016 | 6.4 years | 6,383 | Person | UK | England | >18 | Yes |
| Huang et al. [73] | 2021 | Cohort | Schizophrenia | All cause admissions | Patients with diabetes | | 2002–2013 | 11 | 10,604 | Person | Taiwan | National | Not given | Yes |
| | | **Studies of cardiovascular disease and SMI** | | | | | | | | | | | | |
| Attar et al. [48] | 2020 | Cohort | Schizophrenia | Major adverse cardiac event long-term readmissions | Patients with acute myocardial infarction | | 2000–2018 | 5 years | 286,333 | Person | Sweden | National | >18 | No |
| Chamberlain et al. [49] | 2017 | Cohort | Multimorbidity | All-cause long-term readmissions | Patients with atrial fibrillation | Schizophrenia is one of many risk factors considered | 2000–2014 | Up to 14 years | 2,860 | Person | US | One region | >18 | No |
| Sayers et al. [50] | 2007 | Cross sectional | Psychosis, bipolar disorders, and other MH conditions | All-cause long-term readmissions | Medicare registered patient with congestive heart failure | | 1999 | 1 year | 21,429 | Person | US | National | 65+ | No |
| Shah et al. [51] | 2018 | Cross sectional | Risk factors and comorbidities | All-cause 30-day readmissions | Patients with non-acute myocardial infarction cardiogenic shock | Psychosis is one of many risk factors considered | 2013–2014 | 30 days | 24,665 | Person | US | Multiple regions | >16 | No |
| Pham et al. [52] | 2019 | Cross sectional | Risk factors and comorbidities | All-cause and heart failure 7- and 30-day unplanned readmissions | Medicare registered patient with heart failure | Psychosis is one of many risk factors considered | 2014 | 30 days | 234,298 | Admissions | US | Multiple regions | >65 | No |
| Chamberlain et al. [53] | 2018 | Cross sectional | Risk factors and comorbidities | Heart failure 30-day readmissions | Patients with heart failure | Psychosis is one of many risk factors considered | 2006–2011 | 30 days | 1,007,807 | Person | US | Multiple regions | Not given | No |
| Shah et al. [54] | 2018 | Cross sectional | Risk factors and comorbidities | All cause 31-day readmissions | Patients with Takotsubo cardiomyopathy | Psychosis is one of many risk factors considered | 2013–2014 | 31 days | 5,997 | Person | US | Multiple regions | >18 | No |
| Shah et al. [55] | 2018 | Cross sectional | Risk factors and comorbidities | All-cause 30-day readmissions | Patients with acute myocardial infarction and cardiogenic shock | Psychosis is one of many risk factors considered | 2013–2014 | 30 days | 26,016 | Person | US | Multiple regions | >16 | No |
| Jorgensen et al. [56] | 2017 | Cohort | Schizophrenia | All-cause 28-day readmissions | Patients with heart failure | | 2004–2013 | 28 days | 36,718 | Person | Denmark | National | >18 | No |
| Ahmedani et al. [57] | 2015 | Cohort | Bipolar disorders, schizophrenia-spectrum disorders, other psychoses, and other MH conditions | All-cause 30-day readmissions | Patients with heart failure or myocardial infarction | | 2009–2011 | 30 days | 123,921 | Admissions | US | Multiple regions | >18 | No |
| Coffey et al. [58] | 2012 | Cross sectional | Risk factors and comorbidities | Congestive heart failure 30-day readmissions | Patients with congestive heart failure | Psychosis is one of many risk factors considered | 2006 | 30 days | | Admissions | US | Multiple regions | >18 | No |
| Lu et al. [59] | 2017 | Cohort | Schizophrenia, bipolar mood disorder, and other MH conditions | Heart failure 30-day and long-term readmissions | African American patients with heart failure | | 2010–2013 | 30 days / ave 3.2 years | 611 | Person | US | Single site | >20 | No |
| Kallio et al. [69] | 2022 | Cohort | Schizophrenia | Stroke and myocardial infarction long term readmissions | Patients with coronary artery disease who underwent coronary artery bypass grafting surgery | | 2004–2018 | Up to 10 years | 29,220 | Person | Finland | Multiple sites | Not given | Yes |
| Fleetwood et al. [71] | 2021 | Cohort | Schizophrenia and bipolar disorder | Stroke and myocardial infarction long term readmissions | Patients hospitalised with myocardial infarction | | 1999–2018 | Up to 20 years | 184,134 | Person | UK | Scotland | >18 | No |
| Ghani et al. [72] | 2021 | Cohort | SMI | All-cause 30-day emergency readmissions | Patients who underwent vascular surgery | | 2007–2018 | 30 days | 8,973 | Person | UK | One region | >18 | No |
| Fleetwood et al. [70] | 2021 | Cohort | Schizophrenia and bipolar disorder | Stroke and myocardial infarction long term readmissions | Patients hospitalised with stroke | | 1991–2018 | Up to 28 years | 169,923 | Person | UK | Scotland | >18 | No |
| Paredes et al. [75] | 2020 | Cohort | SMI | All-cause 30-day readmissions | Medicare registered patients who underwent coronary artery bypass grafting surgery | | 2013–2017 | 30 days | 118,837 | Person | US | National | >65 | No |

*(Continued)*

**Table 1.** (Continued)

| Authors | Pub year | Study design | Exposure | Outcome | Population | Notes | Study period | Follow up | Pop size | Unit of measure | Country | Area | Age | Matched |
|---|---|---|---|---|---|---|---|---|---|---|---|---|---|---|
| Sreenivasan et al. [76] | 2022 | Cohort | Bipolar disorder and schizophrenia or other psychotic illnesses | All-cause 30-day readmissions | Patients hospitalised with myocardial infarction | | 2016–2017 | 30 days | 904,575 | Person | US | National | >18 | No |
| Andres et al. [77] | 2012 | Cross sectional | Schizophrenia | Long-term readmission for myocardial infarction | Patients hospitalised with myocardial infarction | | 2000–2007 | 8 years | 19,016 | Person | Spain | One region | >15 | No |
| **Studies of COPD and SMI** | | | | | | | | | | | | | | |
| Buhr et al. [60] | 2019 | Cross sectional | Charlson and Elixhauser indicies | All-cause 30-day readmissions | Patients with COPD | Psychosis included in the Elixhauser index | 2010–2016 | 30 days | 1,622,983 | Admissions | US | National | >40 | No |
| Jorgensen et al. [61] | 2018 | Cohort | Schizophrenia | All-cause 30-day readmissions | Patients with COPD | | 2008–2013 | 30 days | 211,868 | Person | Denmark | National | >30 | No |
| Lau et al. [62] | 2017 | Cross sectional | Risk factors and comorbidities | COPD 30-day readmissions | Patients with COPD | Psychosis is one of many risk factors considered | 2006–2011 | 30 days | 597,502 | Person | US | Multiple regions | >40 | No |
| Singh et al. [63] | 2016 | Cohort | Psychosis, and other MH conditions | All-cause 30-day readmissions | Medicare registered patients with COPD | | 2001–2011 | 30 days | 135,498 | Admissions | US | National | >66 | No |
| **Studies of cancer, liver disease or multiple diseases and SMI** | | | | | | | | | | | | | | |
| Basta et al. [64] | 2016 | Cohort | Risk factors and comorbidities | Complicated lymphedema long-term readmissions | Women who had undergone breast cancer related mastectomy /lumpectomy | Psychosis is one of many risk factors considered | 2007–2012 | 2 years | 56,075 | Person | US | Multiple regions | >18 | No |
| Kashyap et al. [68] | 2021 | Cohort | Bipolar and psychoses | All-cause 30-day ED attendance | Medicare registered patients with gastrointestinal malignancies in the last 30 days of life | | 2004–2014 | 30 days | 110,325 | Person | US | National | >66 | No |
| Ratcliff et al. [74] | 2021 | Cohort | Bipolar disorder and psychoses | All cause 90-day readmissions | Veterans Affairs registered patients who underwent surgery for colorectal cancer | | Not given | 90 days | 50,611 | Person | US | National | Not given | No |
| Huckans et al. [65] | 2010 | Cohort | Schizophrenia | All-cause readmissions during anti-viral therapy | Veterans Affairs registered patients with hepatitis C | | 1998–2006 | During antiviral therapy | 60 | Person | US | Multiple regions | Mean 50 | Yes |
| Davydow et al. [66] | 2016 | Cohort | SMI | ACSC admissions | General population | Table 1 provides unadjusted effect for patients with underlying cardiovascular disease, diabetes, liver disease and cancer | 1999–2013 | 14 years | 5,945,540 | Person | Denmark | National | >18 | No |
| Guo et al. [67] | 2008 | Cohort | Risk factors and comorbidities | All-cause admissions and ED attendances | Commercially insured patients with bipolar disorder | Diabetes, COPD and heart disease are some of many risk factors considered | 1998–2002 | Up to 5 years | 67,862 | Person | US | Multiple regions | Mean 37.1 | No |

analyses had differing NOS for analyses presenting ORs and HRs (S1 and S2 Tables). Funnel plots for all analyses presenting ORs (Egger's test: p = 0.3733, S1 Fig) and risk ratios (Egger's test: p = 0.2809, S1 Fig) were not suggestive of publication bias, however the funnel plot for analyses presenting HRs was asymmetrical (Egger's test: p<0.0001, S1 Fig).

## Hospital utilisation in people with SMI, comparing people with or without physical LTCs

Nine analyses from three studies [39, 46, 67] investigated the impact of diabetes (n = 5), cardio-vascular disease (n = 2) and COPD (n = 2) on hospitalisation in a patient population with pre-

**Table 2. Description of analyses.**

| Authors | Year | Baseline condition | Exposure | Utilisation | Utilisation type | NOS score | Adjusted for age and sex | Adjusted for physical comorbidities | Adjusted for prior utilisation | Effect measure | Effect size | 95%CI/p-value | Included in meta-analysis |
|---|---|---|---|---|---|---|---|---|---|---|---|---|---|
| **The effect of diabetes on hospital utilisation in patients with SMI** | | | | | | | | | | | | | |
| Yan et al. [39] | 2019 | Schizophrenia | Diabetes T2 | Inpatient | All cause | 9 | Yes | Yes | Yes | aOR | 1.19 | 1.05–1.36 | NA |
| Shim et al. [46] | 2014 | Schizophrenia | Diabetes T1/T2 | ED | All cause | 4 | No | No | No | OR | 1.46 | 1.41–1.51 | NA |
| Yan et al. [39] | 2019 | Bipolar | Diabetes T2 | Inpatient | All cause | 9 | Yes | Yes | Yes | aOR | 1.23 | 1.13–1.34 | NA |
| Guo et al. [67] | 2008 | Bipolar | Diabetes | Inpatient | All cause | 6 | Yes | Yes | No | aRR | 1.44 | 1.36–1.52 | NA |
| Guo et al. [67] | 2008 | Bipolar | Diabetes | ED | All cause | 6 | Yes | Yes | No | aRR | 1.17 | 1.08–1.25 | NA |
| **The effect of cardiovascular disease on hospital utilisation in patients with SMI** | | | | | | | | | | | | | |
| Guo et al. [67] | 2008 | Bipolar | Ischemic heart disease | Inpatient | All cause | 6 | Yes | Yes | No | aRR | 1.89 | 1.78–2.02 | NA |
| Guo et al. [67] | 2008 | Bipolar | Ischemic heart disease | ED | All cause | 6 | Yes | Yes | No | aRR | 1.67 | 1.53–1.81 | NA |
| **The effect of COPD on hospital utilisation in patients with SMI** | | | | | | | | | | | | | |
| Guo et al. [67] | 2008 | Bipolar | COPD | Inpatient | All cause | 6 | Yes | Yes | No | aRR | 1.94 | 1.81–2.06 | NA |
| Guo et al. [67] | 2008 | Bipolar | COPD | ED | All cause | 6 | Yes | Yes | No | aRR | 1.61 | 1.47–1.76 | NA |
| **The effect of SMI on hospital utilisation in patients with diabetes** | | | | | | | | | | | | | |
| Stockbridge et al. [31] | 2019 | Diabetes T1/T2 | Bipolar | Inpatient | Diabetes | 7 | Yes | Yes | No | aOR | 0.99 | 0.78–1.25 | Yes |
| Druss et al. [35] | 2012 | Diabetes T1/T2 | Bipolar | Inpatient | ACSC | 7 | Yes | Yes | No | aOR | 1.03 | 0.98–1.09 | Yes |
| Leung et al. [36] | 2011 | Diabetes T2 | Bipolar | Inpatient | Diabetes | 7 | Yes | No | Yes | aOR | 1.07 | 0.91–1.26 | Yes |
| Chen et al. [40] | 2012 | Diabetes T1/T2 | Psychosis | 30-day | All cause | 8 | Yes | Yes | Yes | aOR | 1.15 | 1.03–1.29 | Yes |
| Stockbridge et al. [31] | 2019 | Diabetes T1/T2 | Schizophrenia | Inpatient | Diabetes | 7 | Yes | Yes | No | aOR | 1.61 | 1.29–2.01 | Yes |
| Goueslard et al. [33] | 2018 | Diabetes T1 | Schizophrenia | Inpatient | Diabetes | 6 | Yes | Yes | No | aOR | 2.21 | 1.69–2.88 | Yes |
| Druss et al. [35] | 2012 | Diabetes T1/T2 | Schizophrenia | Inpatient | ACSC | 7 | Yes | Yes | No | aOR | 1.26 | 1.21–1.30 | Yes |
| Leung et al. [36] | 2011 | Diabetes T2 | Schizophrenia | Inpatient | Diabetes | 7 | Yes | No | Yes | aOR | 0.75 | 0.63–0.89 | Yes |
| Guerrero Fernandez de Alba et al. [41] | 2020 | Diabetes T2 | Schizophrenia | Inpatient | All cause | 6 | Yes | Yes | No | aOR | 1.40 | 1.18–1.66 | Yes |
| Guerrero Fernandez de Alba et al. [41] | 2020 | Diabetes T2 | Schizophrenia | Inpatient | Diabetes | 6 | Yes | Yes | No | aOR | 1.25 | 0.55–2.82 | Yes |
| Guerrero Fernandez de Alba et al. [41] | 2020 | Diabetes T2 | Schizophrenia | ED | All cause | 6 | Yes | Yes | No | aOR | 1.28 | 1.11–1.47 | Yes |
| Kurdyak et al. [45] | 2017 | Diabetes T1/T2 | Schizophrenia | ED | Diabetes | 6 | Yes | Yes | No | aOR | 1.34 | 1.28–1.41 | Yes |
| Kurdyak et al. [45] | 2017 | Diabetes T1/T2 | Schizophrenia | ED | All cause[a] | 6 | Yes | Yes | No | aOR | 1.72 | 1.68–1.77 | Yes |
| Kurdyak et al. [45] | 2017 | Diabetes T1/T2 | Schizophrenia | Inpatient | Diabetes | 6 | Yes | Yes | No | aOR | 1.36 | 1.28–1.43 | Yes |
| Kurdyak et al. [45] | 2017 | Diabetes T1/T2 | Schizophrenia | Inpatient | All cause[a] | 6 | Yes | Yes | No | aOR | 1.85 | 1.79–1.92 | Yes |
| Helmer et al. [30] | 2020 | Diabetes T1/T2 | SMI | Inpatient | ACSC | 7 | Yes | Yes | No | aOR | 1.00 | 0.94–1.07 | Yes |
| Chwastiak et al. [42] | 2014 | Diabetes T1/T2 | SMI | 30-day | All cause[a] | 8 | Yes | Yes | Yes | aOR | 1.24 | 1.07–1.44 | Yes |
| Wang et al. [78] | 2021 | Diabetes T2 | SMI | Inpatient | All cause[a] | 9 | Yes | Yes | Yes | aOR | 1.36 | 1.13–1.65 | Yes |
| Cramer et al. [38] | 2010 | Diabetes T1/T2 | Psychosis | Inpatient | All cause | 5 | No | Yes | No | aOR | 2.15 | 1.18–3.92 | Yes, but also excluded as does not adjusted for age and sex |
| Helmer et al. [30] | 2020 | Diabetes T1/T2 | SMI | Inpatient | Chronic ACSC | 7 | Yes | Yes | No | aOR | 0.88 | 0.82–0.96 | No: subset of all ACSC |
| Helmer et al. [30] | 2020 | Diabetes T1/T2 | SMI | Inpatient | Acute ACSC | 7 | Yes | Yes | No | aOR | 1.21 | 1.11–1.31 | No: subset of all ACSC |
| Egglefield et al. [29] | 2020 | Diabetes T1/T2 | Schizophrenia | Inpatient | Diabetes | 4 | No | No | No | OR[d] | 1.69 | 1.54–1.86 | No: unadjusted |
| Krein et al. [44] | 2006 | Diabetes T1/T2 | SMI | Inpatient | All cause | 4 | No | No | No | OR | 2.80 | 2.67–2.94 | No: unadjusted |
| Shim et al. [46] | 2014 | Diabetes T1/T2 | Schizophrenia | ED | Diabetes | 4 | No | No | No | OR | 1.17 | 1.12–1.21 | No: unadjusted |
| Shim et al. [46] | 2014 | Diabetes T1/T2 | Schizophrenia | ED | All cause[a] | 4 | No | No | No | OR[d] | 1.30 | 1.25–1.34 | No: unadjusted |

*(Continued)*

**Table 2.** (Continued)

| Authors | Year | Baseline condition | Exposure | Utilisation | Utilisation type | NOS score | Adjusted for age and sex | Adjusted for physical comorbidities | Adjusted for prior utilisation | Effect measure | Effect size | 95%CI/p-value | Included in meta-analysis |
|---|---|---|---|---|---|---|---|---|---|---|---|---|---|
| Tsai et al. [32] | 2019 | Diabetes T1/T2 | Bipolar | Inpatient | Diabetes | 8 | Yes | Yes | No | aHR | 1.41 | 1.15–1.71 | Yes |
| Mai et al. [37] | 2011 | Diabetes T1/T2 | Affective psychosis | Inpatient | Diabetes | 8 | Yes | Yes | No | aHR[e] | 1.22 | 1.15–1.30 | Yes |
| Edwards et al. [34] | 2014 | Diabetes T1/T2 | Psychosis | Inpatient | ACSC | 6 | Yes | Yes | No | aHR | 1.01 | 0.98–1.04 | Yes |
| Mai et al. [37] | 2011 | Diabetes T1/T2 | Other psychosis | Inpatient | Diabetes | 8 | Yes | Yes | No | aHR[e] | 1.18 | 1.10–1.27 | Yes |
| Mai et al. [37] | 2011 | Diabetes T1/T2 | Schizophrenia | Inpatient | Diabetes | 8 | Yes | Yes | No | aHR[e] | 1.06 | 0.94–1.20 | Yes |
| Becker et al. [43] | 2011 | Diabetes T1/T2 | Schizophrenia | Inpatient or ED | Diabetes | 8 | Yes | Yes | Yes | aHR | 1.68 | 1.34–2.10 | Yes |
| Chwastiak et al. [42] | 2014 | Diabetes T1/T2 | SMI | Inpatient | All cause[a] | 7 | Yes | Yes | Yes | aHR | 1.14 | 1.05–1.23 | Yes |
| Goueslard et al. [33] | 2018 | Diabetes T1 | Schizophrenia | Inpatient | Diabetes | 6 | Yes | Yes | No | aHR | 2.13 | 1.69–2.69 | Yes, but also excluded as an outlier |
| Stockbridge et al. [31] | 2019 | Diabetes T1/T2 | Bipolar | Inpatient | Diabetes | 7 | Yes | Yes | No | aRR | 1.34 | 0.78–2.31 | No: RR |
| Stockbridge et al. [31] | 2019 | Diabetes T1/T2 | Schizophrenia | Inpatient | Diabetes | 7 | Yes | Yes | No | aRR | 1.41 | 0.94–2.12 | No: RR |
| Huang et al. [73] | 2021 | Diabetes T2 | Schizophrenia | Inpatient | All cause[a] | 7 | No | No | No | Average number of admissions | 1.09 vs 0.92 | p = 0.001 | No: Average utilisation |
| Sullivan et al. [47] | 2006 | Diabetes T1/T2 | SMI | Admission ratio | Diabetes | 6 | Yes | No | No | aOR | 0.77 | 0.45–1.33 | No: Admission ratio |
| **The effect of SMI on hospital utilisation in patients with cardiovascular disease** | | | | | | | | | | | | | |
| Shah et al. [51] | 2018 | Cardiogenic shock (no AMI) | Psychosis | 30-day | All cause | 8 | Yes | Yes | No | aOR | 0.90 | 0.78–1.05 | Yes |
| Pham et al. [52] | 2019 | Heart failure | Psychosis | 30-day | All cause | 7 | Yes | Yes | No | aOR | 1.11 | 1.04–1.18 | Yes |
| Pham et al. [52] | 2019 | Heart failure | Psychosis | 30-day | Cardiovascular | 7 | Yes | Yes | No | aOR | 1.02 | 0.93–1.13 | Yes |
| Chamberlain et al. [53] | 2018 | Congestive heart failure | Psychosis | 30-day | Cardiovascular | 8 | Yes | Yes | No | aOR | 1.07 | 1.01–1.12 | Yes |
| Chamberlain et al. [53] | 2018 | Congestive heart failure | Psychosis | 30-day | Cardiovascular | 8 | Yes | Yes | No | aOR | 1.08 | 1.00–1.16 | Yes |
| Shah et al. [54] | 2018 | Takotsubo cardiomyopathy | Psychosis | 30-day | All cause | 8 | Yes | Yes | No | aOR | 1.90 | 1.36–2.66 | Yes |
| Shah et al. [55] | 2018 | Cardiogenic shock (with AMI) | Psychosis | 30-day | All cause | 8 | Yes | Yes | No | aOR | 1.14 | 0.97–1.35 | Yes |
| Coffey et al. [58] | 2012 | Congestive heart failure | Psychosis | 30-day | Cardiovascular | 7 | Yes | Yes | No | aOR | 1.16 | p<0.001 | Yes |
| Jorgensen et al. [56] | 2017 | Heart failure | Schizophrenia | 30-day | All cause[a] | 9 | Yes | Yes | No | aOR | 1.77 | 0.79–3.92 | Yes |
| Ghani et al. [72] | 2021 | Vascular surgery | SMI | 30-day | All cause[c] | 6 | Yes | No | Yes | aOR | 2.02 | 1.10–3.70 | Yes |
| Paredes et al. [75] | 2020 | CABG surgery | SMI | 30-day | All cause | 7 | Yes | Yes | No | aOR[e] | 2.28 | 2.10–2.46 | Yes |
| Pham et al. [52] | 2019 | Heart failure | Psychosis | 7-day | All cause | 7 | Yes | Yes | No | aOR | 1.10 | 1.00–1.22 | No: 7-day readmission |
| Pham et al. [52] | 2019 | Heart failure | Psychosis | 7-day | Cardiovascular | 7 | Yes | Yes | No | aOR | 1.04 | 0.87–1.23 | No: 7-day readmission |
| Ahmedani et al. [57] | 2015 | Heart failure | Schizophrenia | 30-day | All cause | 6 | No | No | No | OR[d] | 1.06 | 0.78–1.44 | No: unadjusted |
| Ahmedani et al. [57] | 2015 | MI | Schizophrenia | 30-day | All cause | 6 | No | No | No | OR[d] | 1.55 | 0.69–3.45 | No: unadjusted |
| Ahmedani et al. [57] | 2015 | Heart failure | Bipolar | 30-day | All cause | 6 | No | No | No | OR[d] | 1.25 | 1.05–1.50 | No: unadjusted |
| Ahmedani et al. [57] | 2015 | MI | Bipolar | 30-day | All cause | 6 | No | No | No | OR[d] | 0.98 | 0.61–1.58 | No: unadjusted |
| Ahmedani et al. [57] | 2015 | Heart failure | Other psychoses | 30-day | All cause | 6 | No | No | No | OR[d] | 1.70 | 1.40–2.07 | No: unadjusted |
| Andres et al. [77] | 2012 | MI | Schizophrenia | Inpatient | MI | 6 | No | No | No | OR[d] | 0.83 | 0.25–2.81 | No: unadjusted |
| Sreenivasan et al. [76] | 2022 | MI | Psychosis | 30-day | All cause | 8 | Yes | Yes | No | aHR | 1.56 | 1.43–1.69 | Yes |
| Lu et al. [59] | 2017 | Heart failure | Bipolar | Inpatient | Cardiovascular | 6 | Yes | Yes | No | aHR | 2.08 | 1.05–4.11 | Yes, but also excluded as an outlier |

*(Continued)*

**Table 2.** (Continued)

| Authors | Year | Baseline condition | Exposure | Utilisation | Utilisation type | NOS score | Adjusted for age and sex | Adjusted for physical comorbidities | Adjusted for prior utilisation | Effect measure | Effect size | 95%CI/p-value | Included in meta-analysis |
|---|---|---|---|---|---|---|---|---|---|---|---|---|---|
| Fleetwood et al. [71] | 2021 | MI | Bipolar | Inpatient | MI or stroke | 8 | Yes | No | No | aHR | 1.40 | 1.20–1.62 | Yes |
| Fleetwood et al. [70] | 2021 | Stroke | Bipolar | Inpatient | MI or stroke | 8 | Yes | No | No | aHR | 1.14 | 1.01–1.28 | Yes |
| Sreenivasan et al. [76] | 2022 | MI | Bipolar | 30-day | All cause | 8 | Yes | Yes | No | aHR | 1.32 | 1.19–1.45 | Yes |
| Lu et al. [59] | 2017 | Heart failure | Bipolar | 30-day | Cardiovascular | 7 | Yes | Yes | No | aHR | 3.44 | 1.19–10.00 | Yes, but also excluded as an outlier |
| Attar et al. [48] | 2020 | MI | Schizophrenia | Inpatient | Re-infarction | 8 | Yes | Yes | Yes | aHR | 1.29 | 0.77–2.13 | Yes |
| Chamberlain et al [49] | 2017 | Atrial fibrillation | Schizophrenia | Inpatient | All cause | 7 | Yes | Yes | No | aHR | 1.22 | 0.98–1.52 | Yes |
| Lu et al. [59] | 2017 | Heart failure | Schizophrenia | Inpatient | Cardiovascular | 6 | Yes | Yes | No | aHR | 2.33 | 1.51–3.61 | Yes, but also excluded as an outlier |
| Lu et al. [59] | 2017 | Heart failure | Schizophrenia | 30-day | Cardiovascular | 7 | Yes | Yes | No | aHR | 4.92 | 2.49–9.71 | Yes, but also excluded as an outlier |
| Fleetwood et al. [71] | 2021 | MI | Schizophrenia | Inpatient | MI or stroke | 8 | Yes | No | No | aHR | 1.46 | 1.29–1.65 | Yes |
| Fleetwood et al. [70] | 2021 | Stroke | Schizophrenia | Inpatient | MI or stroke | 8 | Yes | No | No | aHR | 1.21 | 1.10–1.34 | Yes |
| Fleetwood et al. [71] | 2021 | MI | Schizophrenia | Inpatient | MI | 8 | Yes | No | No | aHR | 1.42 | 1.24–1.63 | No: Population included in other outcome |
| Fleetwood et al. [71] | 2021 | MI | Bipolar | Inpatient | MI | 8 | Yes | No | No | aHR | 1.34 | 1.13–1.58 | No: Population included in other outcome |
| Fleetwood et al. [70] | 2021 | Stroke | Schizophrenia | Inpatient | Stroke | 8 | Yes | No | No | aHR | 1.24 | 1.11–1.38 | No: Population included in other outcome |
| Fleetwood et al. [70] | 2021 | Stroke | Bipolar | Inpatient | Stroke | 8 | Yes | No | No | aHR | 1.17 | 1.03–1.32 | No: Population included in other outcome |
| Attar et al. [48] | 2020 | MI | Schizophrenia | Inpatient | Stroke | 8 | Yes | Yes | Yes | aHR | 1.72 | 1.00–2.98 | No: Population included in other outcome |
| Attar et al. [48] | 2020 | MI | Schizophrenia | Inpatient | Heart failure | 8 | Yes | Yes | Yes | aHR | 1.39 | 1.04–1.86 | No: Population included in other outcome |
| Kallio et al. [69] | 2022 | Coronary artery disease and CABG | Schizophrenia | Inpatient | MI | 6 | No | No | No | HR | 1.86 | 1.25–2.78 | No: unadjusted |
| Kallio et al. [69] | 2022 | Coronary artery disease and CABG | Schizophrenia | Inpatient | Stroke | 6 | No | No | No | HR | 0.91 | 0.50–1.66 | No: unadjusted |
| Sayers et al. [50] | 2007 | Heart failure | Psychosis | Inpatient | All cause | 7 | Yes | Yes | No | Predicted increase | 0.30 | p<0.001 | No: predicted increase |
| Sayers et al. [50] | 2007 | Heart failure | Bipolar | Inpatient | All cause | 7 | Yes | Yes | No | Predicted increase | 0.38 | p = 0.001 | No: predicted increase |
| Davydow et al. [66] | 2016 | MI | SMI | Inpatient | ACSC | 5 | No | No | No | RR[d] | 1.41 | 1.36–1.47 | No: RR |
| Davydow et al. [66] | 2016 | CHF | SMI | Inpatient | ACSC | 5 | No | No | No | RR[d] | 1.19 | 1.15–1.22 | No: RR |
| Davydow et al. [66] | 2016 | Cerebrovascular disease | SMI | Inpatient | ACSC | 5 | No | No | No | RR[d] | 1.47 | 1.43–1.52 | No: RR |
| **The effect of SMI on hospital utilisation in patients with COPD** | | | | | | | | | | | | | |
| Lau et al. [62] | 2017 | COPD | Psychosis | 30-day | COPD | 8 | Yes | Yes | No | aOR | 1.19 | 1.13–1.25 | Yes |
| Lau et al. [62] | 2017 | COPD | Psychosis | 30-day | COPD | 8 | Yes | Yes | No | aOR | 1.16 | 1.08–1.24 | Yes |
| Singh et al. [63] | 2016 | COPD | Psychosis | 30-day | All cause | 6 | Yes | No | No | aOR | 1.18 | 1.10–1.27 | Yes |
| Jorgensen et al. [61] | 2018 | COPD | Schizophrenia | 30-day | All cause | 8 | Yes | Yes | No | aOR | 1.08 | 0.92–1.28 | Yes |
| Buhr et al. [60] | 2019 | COPD | Psychosis | 30-day | All cause | 5 | No | No | No | OR[d] | 1.27 | 1.25–1.29 | No: unadjusted |
| **The effect of SMI on inpatient admissions in liver disease patients** | | | | | | | | | | | | | |
| Huckans et al. [65] | 2010 | HCV | Schizophrenia | Inpatient | All cause[a] | 5 | No | No | No | OR | 5.80 | 0.63–53.01 | No: unadjusted |

(*Continued*)

**Table 2.** (Continued)

| Authors | Year | Baseline condition | Exposure | Utilisation | Utilisation type | NOS score | Adjusted for age and sex | Adjusted for physical comorbidities | Adjusted for prior utilisation | Effect measure | Effect size | 95%CI/p-value | Included in meta-analysis |
|---|---|---|---|---|---|---|---|---|---|---|---|---|---|
| Huckans et al. [65] | 2010 | HCV | Schizophrenia | ED | All cause[a] | 5 | No | No | No | OR | 3.27 | 0.77–13.83 | No: unadjusted |
| Davydow et al. [66] | 2016 | Liver disease | SMI | Inpatient | ACSC | 5 | No | No | No | RR[d] | 1.53 | 1.45–1.61 | No: unadjusted |
| **The effect of SMI on inpatient admissions in cancer patients** | | | | | | | | | | | | | |
| Davydow et al. [66] | 2016 | Cancer | SMI | Inpatient | ACSC | 5 | No | No | No | RR[d] | 1.54 | 1.48–1.60 | No: unadjusted |
| Basta et al. [64] | 2016 | Breast cancer related mastectomy/ lumpectomy | Psychosis | Inpatient | Cancer | 8 | No[b] | Yes | No | aOR | 2.15 | 1.51–3.06 | No: limited comparison |
| Kashyap et al. [68] | 2021 | Gastrointestinal malignancies | Bipolar | ED | All cause end of life | 8 | Yes | Yes | No | aOR | 1.12 | 1.01–1.24 | No: limited comparison |
| Kashyap et al. [68] | 2021 | Gastrointestinal malignancies | Psychosis | ED | All cause end of life | 8 | Yes | Yes | No | aOR | 0.98 | 0.85–1.12 | No: limited comparison |
| Ratcliff et al. [74] | 2021 | Surgery for colorectal cancer | Bipolar | 90-day | All cause | 6 | No | No | No | OR[d] | 1.24 | 1.04–1.47 | No: unadjusted |
| Ratcliff et al. [74] | 2021 | Surgery for colorectal cancer | Psychosis | 90-day | All cause | 6 | No | No | No | OR[d] | 1.25 | 1.03–1.52 | No: unadjusted |

a: Excluded psychiatric hospitalisations

b: Adjusted for age and only included females so scored as if adjusted for age and sex

c: emergency admissions

d: calculated from raw data

e: extracted from figure using ImageJ: https://imagej.nih.gov/ij/; COPD: Chronic Obstructive Pulmonary Disease; ED: Emergency Department; OR: odds ratio; HR: hazard ratio; RR: risk ratio; HCV: hepatitis C virus; SMI: severe mental illness; CABG: coronary artery bypass graft; MI: myocardial infarction; ACSC: ambulatory care sensitive condition.

existing schizophrenia (n = 2) or bipolar disorder (n = 7). The outcome was all-cause ED attendances for four studies and all-cause admissions for five. All analyses found a higher risk of hospital utilisation in those with SMI and a physical health condition compared to those with SMI alone (Table 2). The low number and heterogenous study characteristics meant that these studies were deemed unsuitable for meta-analysis.

## Hospital utilisation in people with physical LTCs, comparing people with and without SMI

Ninety-five analyses from 48 studies investigated the impact of SMI diagnosis on hospital utilisation in a patient population with diagnoses of diabetes, cardiovascular disease, COPD, liver disease or cancer.

**Hospital utilisation in people with diabetes, with and without SMI.** Thirty-seven analyses from 20 studies investigated the effect of SMI on hospital utilisation in patients with diabetes. Most analyses included patients diagnosed with either type I or II diabetes mellitus (n = 28; Table 2). Twenty-seven analyses were included in meta-analysis, reasons for exclusions are detailed in Table 2.

The meta-analysis of adjusted OR included 19 analyses from 14 studies (Fig 2). Schizophrenia was the most frequent exposure (11 analyses) and admissions the most frequent outcome (14 analyses; Table 2). The funnel plot of these analyses did not show asymmetry (Egger's test: p = 0.0738, S2 Fig). For patients with diabetes, the pooled OR for hospital utilisation in patients with a diagnosis of any SMI was 1.30 (95%CI: 1.16–1.45) compared to those without an SMI diagnosis, however heterogeneity was high ($I^2$ = 97.8%). When one study which did not control for age was removed [38] the pooled odds ratio was 1.28 (95% confidence interval (CI)

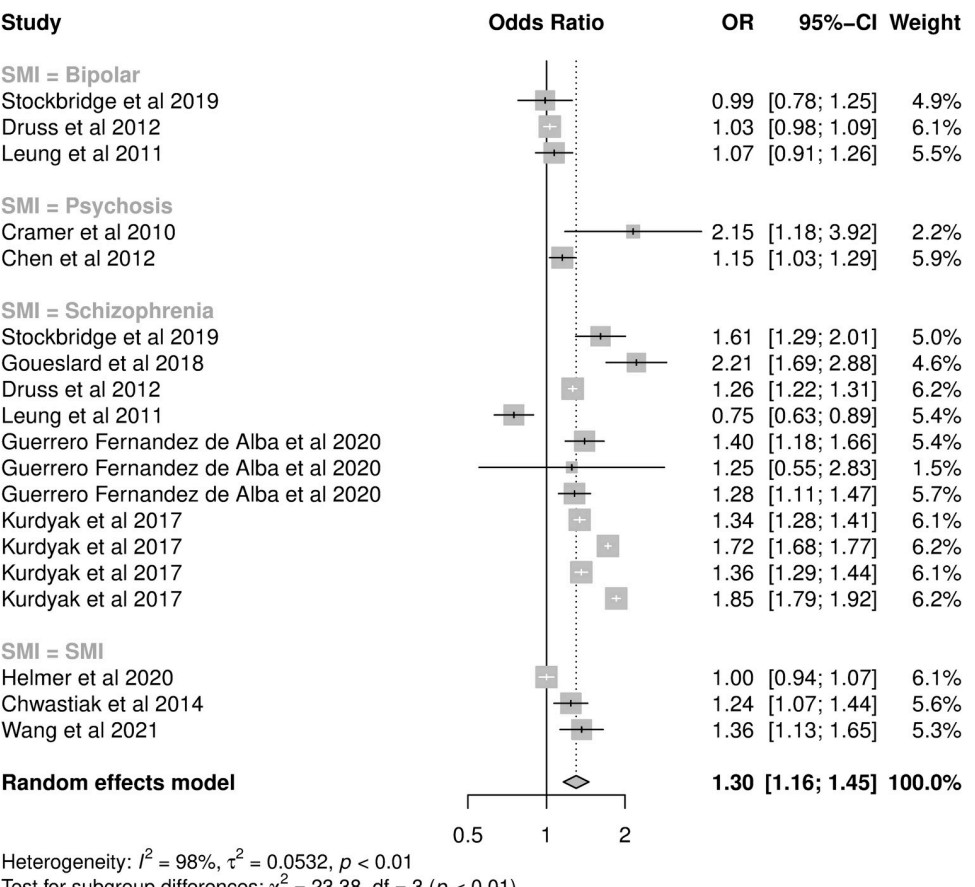

**Fig 2. Forest plot of studies presenting adjusted odds ratios of hospital utilisation in diabetes patients with SMI compared to diabetes patients without SMI.**

1.15–1.44, $I^2$ = 97.9%) In subgroup analysis, the effect size was greater in patients with schizophrenia (OR: 1.42, 95%CI: 1.25–1.60) than patients with other SMI diagnoses, and analyses of all-cause hospitalisations had higher pooled OR (1.43, 95%CI: 1.28–1.60) compared to those reporting ACSC conditions or diabetes-specific hospitalisations (Table 3). Studies performed in the US had a lower pooled OR (1.10, 95%CI: 0.99–1.22) than studies in other countries (Table 3). While the pooled OR for analyses of 30-day readmissions was lower, confidence intervals of all outcome types overlapped (Table 3). Controlling for these variables in meta-regression reduced heterogeneity ($I^2$ = 82.8%).

Fewer studies in populations with diabetes assessed HR (eight analyses from six studies, Fig 3). Seven analyses investigated admissions, while one investigated admissions or ED attendance combined (Table 2). The funnel plot identified one outlier, with a large effect size [33] (S3 Fig). When this outlier was removed, the pooled HR was reduced from 1.26 (1.13–1.41; $I^2$ = 92.7%, Fig 3) to 1.19 (95%CI: 1.08–1.31, $I^2$ = 90.6%). In subgroup analysis, analyses of diabetes admissions had a higher pooled HR (1.25; 95%CI: 1.13–1.37) than all-cause or ACSC admissions studies, while analyses performed in the US had a lower pooled HR (1.07; 95%CI: 0.95–1.20) than studies in other countries (Table 3). Pooled HRs were similar across SMI diagnoses. When controlling for country and type of hospital utilisation in meta-regression, the residual heterogeneity was reduced ($I^2$: 46.5%).

**Table 3. Subgroup analyses of studies of hospital use in people with underlying diabetes, cardiovascular disease and COPD: Comparing those with and without SMI with outliers removed.**

| | | | No. of studies | Pooled effect size (95%CI) of hospital use in people with SMI compared to those without | $I^2$ (%) | p-value for between group differences |
|---|---|---|---|---|---|---|
| **The effect of SMI on hospital use in people with diabetes (OR)** | | | | | | |
| | SMI diagnosis | | | | | <0.0001 |
| | | Bipolar disorder | 3 | 1.03 (0.98–1.08) | 0 | |
| | | Psychosis | 1 | 1.15 (1.03–1.29) | – | |
| | | Schizophrenia | 11 | 1.42 (1.25–1.60) | 97.7 | |
| | | SMI | 3 | 1.17 (0.96–1.44) | 85.9 | |
| | Outcome: service | | | | | 0.2015 |
| | | 30-day readmission | 2 | 1.18 (1.08–1.29) | 0 | |
| | | ED attendance | 3 | 1.44 (1.18–1.77) | 97.8 | |
| | | Inpatient admissions | 13 | 1.26 (1.08–1.47) | 97.9 | |
| | Outcome: Cause | | | | | 0.0225 |
| | | All-cause | 7 | 1.43 (1.28–1.60) | 94.7 | |
| | | Diabetes | 8 | 1.25 (1.08–1.44) | 90.3 | |
| | | Ambulatory care sensitive | 3 | 1.09 (0.93–1.28) | 96.6 | |
| | Country of study | | | | | <0.0001 |
| | | US | 9 | 1.10 (0.99–1.22) | 91.6 | |
| | | Canada | 4 | 1.55 (1.34–1.80) | 98.2 | |
| | | France | 1 | 2.21 (1.69–2.89) | – | |
| | | Spain | 3 | 1.33 (1.19–1.48) | 0 | |
| | | UK | 1 | 1.36 (1.13–1.65) | – | |
| **The effect of SMI on hospital use in people with diabetes (HR)** | | | | | | |
| | SMI diagnosis | | | | | 0.3654 |
| | | Bipolar disorder | 2 | 1.27 (1.12–1.44) | 46.2 | |
| | | Psychosis | 2 | 1.09 (0.93–1.27) | 93.2 | |
| | | Schizophrenia | 2 | 1.32 (0.85–2.07) | 91.8 | |
| | | SMI | 1 | 1.14 (1.05–1.23) | – | |
| | Outcome: Cause | | | | | <0.0001 |
| | | All-cause | 1 | 1.14 (1.05–1.23) | – | |
| | | Diabetes | 5 | 1.25 (1.13–1.37) | 73.5 | |
| | | Ambulatory care sensitive | 1 | 1.01 (0.98–1.04) | – | |
| | Country of study | | | | | 0.0016 |
| | | US | 2 | 1.07 (0.95–1.20) | 87.3 | |
| | | Canada | 1 | 1.68 (1.34–2.10) | – | |
| | | Australia | 3 | 1.17 (1.10–1.25) | 46.5 | |
| | | Taiwan | 1 | 1.41 (1.16–1.72) | – | |
| **The effect of SMI on hospital use in people with cardiovascular disease (OR)** | | | | | | |
| | SMI diagnosis | | | | | <0.0001 |

(*Continued*)

**Table 3.** (Continued)

| | | No. of studies | Pooled effect size (95%CI) of hospital use in people with SMI compared to those without | I² (%) | p-value for between group differences |
|---|---|---|---|---|---|
| | Psychosis | 8 | 1.09 (1.02–1.16) | 66.4 | |
| | Schizophrenia | 1 | 1.77 (0.79–3.94) | – | |
| | SMI | 2 | 2.28 (2.11–2.46) | 0 | |
| Outcome: Cause | | | | | 0.0861 |
| | All-cause | 7 | 1.46 (1.03–2.08) | 97.5 | |
| | Cardiovascular disease | 4 | 1.07 (1.04–1.11) | 0 | |
| Country of study | | | | | 0.2259 |
| | US | 9 | 1.22 (1.01–1.48) | 97.5 | |
| | Denmark | 1 | 1.77 (0.79–3.94) | – | |
| | UK | 1 | 2.02 (1.10–3.70) | – | |
| **The effect of SMI on hospital use in people with cardiovascular disease (HR)** | | | | | |
| | SMI diagnosis | | | | 0.0056 |
| | Bipolar | 3 | 1.28 (1.13–1.43) | 62.9 | |
| | Psychosis | 1 | 1.56 (1.44–1.67) | – | |
| | Schizophrenia | 4 | 1.30 (1.15–1.46) | 47.8 | |
| Outcome: Service | | | | | 0.2218 |
| | 30-day readmission | 2 | 1.44 (1.22–1.69) | 53.7 | |
| | Inpatient admissions | 6 | 1.28 (1.17–1.40) | 84.4 | |
| Outcome: Cause | | | | | 0.4218 |
| | All-cause | 3 | 1.39 (1.20–1.60) | 77.0 | |
| | Cardiovascular disease | 5 | 1.29 (1.16–1.43) | 62.5 | |
| Country of study | | | | | 0.7365 |
| | Sweden | 1 | 1.29 (0.78–2.14) | – | |
| | UK | 4 | 1.29 (1.15–1.44) | 71.8 | |
| | US | 3 | 1.39 (1.20–1.60) | 77.0 | |
| **The effect of SMI on hospital use in people with COPD (OR)** | | | | | |
| | SMI diagnosis | | | | 0.3059 |
| | Psychosis | 3 | 1.18 (1.14–1.22) | 0 | |
| | Schizophrenia | 1 | 1.08 (0.92–1.27) | – | |
| Outcome: Cause | | | | | 0.7298 |
| | All-cause | 2 | 1.16 (1.09–1.24) | 0 | |
| | COPD | 2 | 1.18 (1.13–1.23) | 0 | |
| Country of study | | | | | 0.3059 |
| | US | 3 | 1.18 (1.14–1.22) | 0 | |
| | Denmark | 1 | 1.08 (0.92–1.27) | – | |

For studies of both hazard ratios and odds ratios based in the US, there was evidence that pooled effect sizes of hospital utilisation in people with SMI were lower in studies of patients registered in Veteran's Affairs, Medicare or Medicaid, compared to studies of commercially

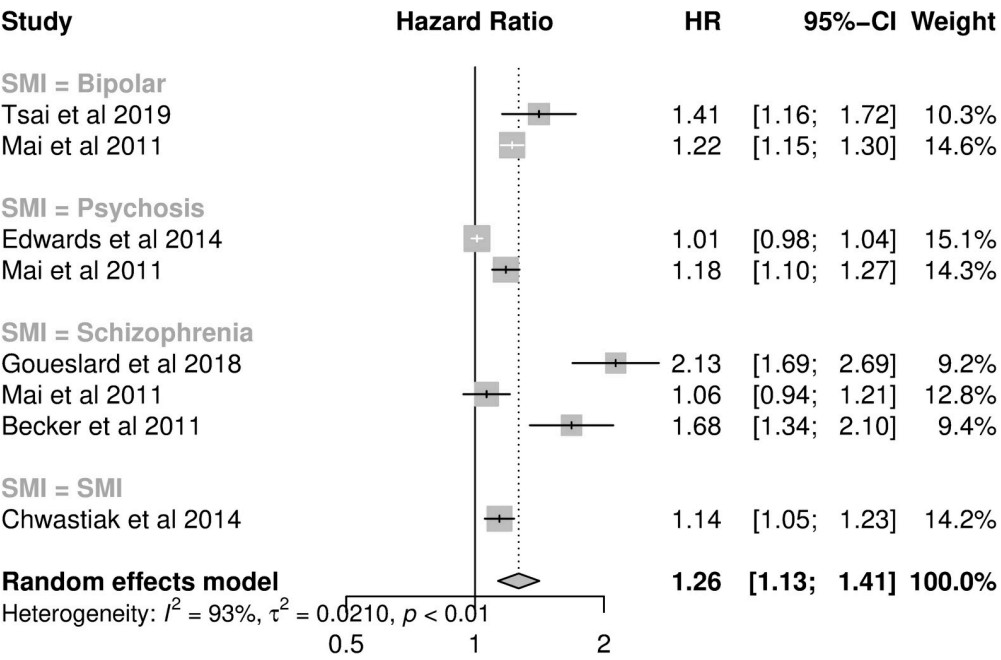

**Fig 3. Forest plot of studies presenting adjusted hazard ratios of hospital utilisation in diabetes patients with SMI compared to diabetes patients without SMI.**

insured people or studies including both state-insured and commercially insured individuals (Table 4).

**Hospitalisation use in people with cardiovascular disease, with and without SMI.**
Forty-four analyses from 20 studies were based in populations with underlying cardiovascular

**Table 4. Subgroup analyses of studies of hospital use in the US in people with underlying diabetes: Comparing those with and without SMI.**

| | | | No. of studies | Pooled effect size (95%CI) of hospital use in people with SMI compared to those without | I$^2$ (%) | p-value for between group differences |
|---|---|---|---|---|---|---|
| **The effect of SMI on hospital use in people with diabetes (OR)** | | | | | | |
| | Study population | | | | | 0.0365 |
| | | Medicaid/ Medicare | 4 | 1.03 (0.86–1.22) | 95.4 | |
| | | Veterans' health | 1 | 1.00 (0.94–1.07) | – | |
| | | Insured | 3 | 1.22 (0.96–1.56) | 80.1 | |
| | | Complete | 1 | 1.24 (1.07–1.44) | – | |
| **The effect of SMI on hospital use in people with diabetes (HR)** | | | | | | |
| | Study population | | | | | 0.005 |
| | | Veterans' health | 1 | 1.01 (0.98–1.04) | | |
| | | Insured | 1 | 1.14 (1.05–1.23) | | |

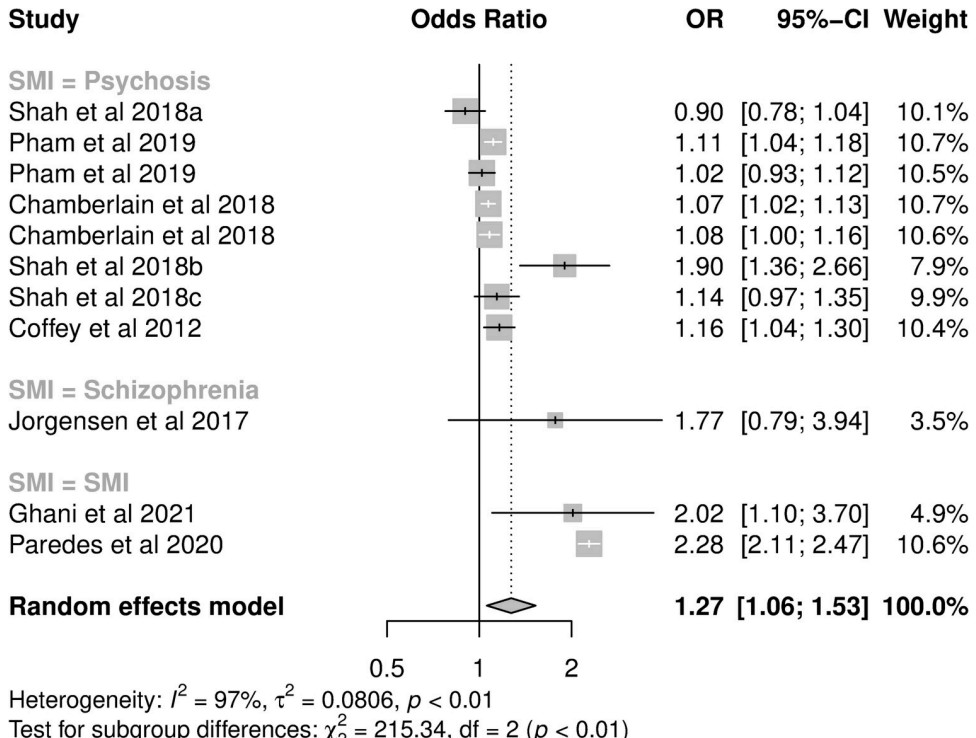

**Fig 4. Forest plot of studies presenting adjusted odds ratios of hospital utilisation in cardiovascular disease patients with SMI compared to cardiovascular disease without SMI.** a: [51], b: [54], c: [55].

disease, the most common of which was heart failure (n = 7, Table 1). Eleven analyses from nine studies providing adjusted ORs for hospital utilisation in people with SMI compared to those without SMI were included in meta-analysis, and twelve analyses from six studies presented adjusted HR. The funnel plot for these analyses did not show asymmetry (meta-analysis of ORs: Egger's test: p = 0.6751, S4 Fig; meta-analysis of HRs: Egger's test: p = 0.1535, S5 Fig), and for ORs did not show any outliers.

For those presenting ORs, all were 30-day readmission studies, and psychosis was the exposure for eight analyses (Table 2). The pooled OR for hospital utilisation in patients with a diagnosis of any SMI was 1.27 (95%CI: 1.06–1.53; I$^2$: 96.9%, Fig 4). In subgroup analysis, pooled OR were not significantly different between cause of hospitalisation or country of study, but did differ by SMI diagnosis (Table 3). The majority of analyses examined broad risk factors for hospitalisation, while only three focused on SMI specifically. Those with SMI as a focus had greater pooled OR (pOR: 2.27, 95%CI: 2.10–2.46 vs. pOR 1.09, 95%CI: 1.02–1.16). Controlling for these variables in meta-regression reduced heterogeneity (I$^2$ = 61.9%).

For those presenting HRs, the pooled HR for hospital utilisation was 1.43 (95%CI: 1.28–1.60, I$^2$: 78.4%, Fig 5). Most analyses investigated inpatient admissions (8/12) and cardiovascular outcomes (n = 9). One study, contributing four analyses, was identified as an outlier (S5 Fig). This study was a small single-site study of African American patients in the US [59]. Removal of this study from the meta-analysis reduced the pooled HR to 1.33 (95%CI: 1.21–1.46, I$^2$: 74.0%). In subgroup analysis, pooled HRs were not significantly different between cause of hospitalisation, hospitalisation type or country of study (Table 3). However, there were differences by SMI diagnosis, and controlling for this did reduce heterogeneity (I$^2$ = 55.13%).

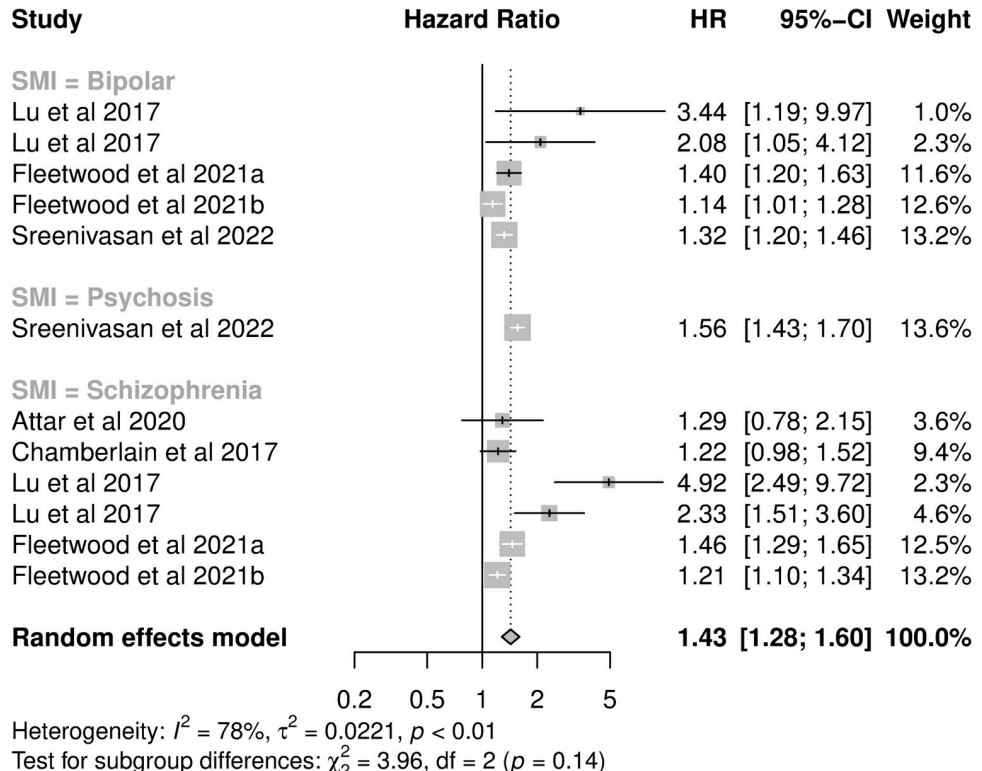

**Fig 5. Forest plot of studies presenting adjusted hazard ratios of hospital utilisation in cardiovascular disease patients with SMI compared to cardiovascular disease without SMI.** a: [71] b: [70].

**Hospitalisation use in people with COPD, with and without SMI.** Five analyses from four studies were in populations with underlying COPD. All five presented ORs for 30-day readmissions in patients with SMI compared to those without SMI, of which four presented adjusted ORs. The funnel plot of these analyses did not show asymmetry of outliers (S6 Fig).

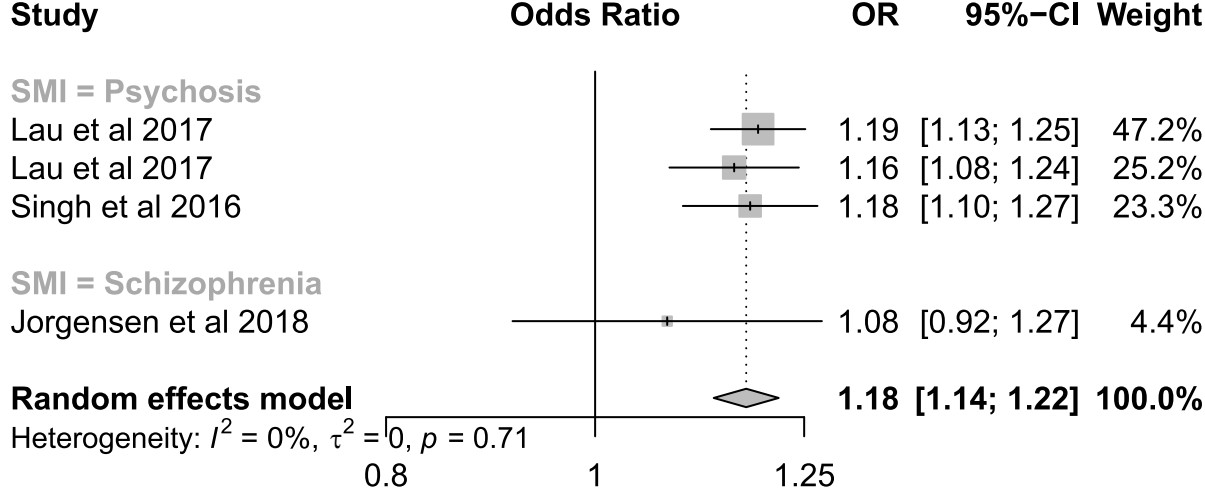

**Fig 6. Forest plot of studies presenting adjusted odds ratios of hospital utilisation in COPD patients with SMI compared to COPD patients without SMI.**

The pooled OR for hospital use in patients with a diagnosis of any SMI was 1.18 (95%CI: 1.14–1.22, $I^2$ = 0%, Fig 6). In subgroup analysis, pooled ORs were not significantly different between cause of hospitalisation, country of study or SMI diagnosis (Table 3).

**Hospitalisation use in people with cancer or liver disease, with and without SMI.** Two studies were identified which considered SMI as an exposure for hospitalisation in people with and without SMI in populations with underlying liver disease and four in populations with underlying cancer (Table 1). Neither of the liver disease studies presented adjusted effect estimates, and both were low quality for the exposures and outcomes considered in this synthesis (NOS score = 5). Huckans et al. [65] found that people with schizophrenia were more likely to attend EDs and have inpatient admissions during hepatitis C treatment than those without schizophrenia, though due to the small population size (n = 60) confidence intervals were wide and included one. Davydow et al. [66] found higher ACSC admissions for in those with liver disease and SMI compared to those with liver disease without SMI (Table 2).

For cancer, two studies presented adjusted effect measures of hospital utilisation. Basta et al. [64] studied readmissions for lymphedema in the two years after breast cancer diagnosis in women. They found that women with a diagnosis of psychosis were at higher risk of readmission (aOR: 2.15, 95%CI: 1.51–3.06). Kashyap et al. [68] found higher utilisation of emergency departments in the 30 days prior to death in those with gastrointestinal malignancies and SMI compared to those with gastrointestinal malignancies alone. Finally, an unadjusted analysis by Ratcliff et al. [74] found higher risk of 90-day readmissions after surgery for colorectal cancer in those with SMI, while Davydow et al. [66] found higher risk of ACSC admissions in those with cancer and SMI compared to those with cancer alone, in unadjusted analysis (Table 2).

*Sensitivity analysis.* In sensitivity analysis, re-running the analysis as a three-level hierarchical model did not result in improved model fit, nor substantial change the pooled OR (1.26, 95%CI: 1.10–1.45) or HR (1.23; 95%CI 1.01–1.50) for studies in people with diabetes and SMI, or the pooled OR (1.34, 95%CI: 1.07–1.69) or HR (1.47. 95%CI: 1.16–1.85) for people with cardiovascular disease and SMI. For COPD, the two analyses from one study included in the meta-analysis were from different populations and so sensitivity analysis was not performed. Only three conference abstracts providing adjusted effect measures for hospitalisation were retrieved. The first was a study of risk factors for 30- and 90-day rehospitalisation following radical cystectomy for bladder cancer. The authors found that people with psychosis had an elevated HR for readmission (aHR: 1.82, p<0.05) [79]. The second was a small study of 373 people with diabetes, which found that those with two or more admissions were more likely to have a diagnosis of schizophrenia (aOR: 4.99, p<0.05) than those with only one admission [80]. Finally, a study of all-cause 30-day readmissions in people with acute ischaemic stroke, found those with SMI we at higher risk (aOR: 1.24, 95%CI: 1.20–1.27) [81].

## Discussion

This review and meta-analysis demonstrates that people with SMI and one of five physical health conditions have consistently higher hospital utilisation than either people with SMI alone or with physical health conditions alone. This is the first systematic review to consider the impact of having SMI and a specific physical health condition on hospital utilisation, allowing a better understanding of the impact of SMI on hospital use in those with underlying physical illness, and highlighting areas for future research.

We found that in people with underlying cardiovascular disease, COPD or diabetes, people with a diagnosis of SMI had higher hospital use compared to those without SMI. This finding is in line with other systematic reviews or meta-analyses [11–13, 17], which consider the

impact of SMI on hospitalisations in the general population, or when controlling for physical health comorbidities. The same appeared to be true for people with cancer and liver disease, though studies presenting adjusted analyses were limited to one study of breast cancer complications [64], and one of end of life emergency department use in people with gastrointestinal malignancies [68]. No studies of liver disease reported adjusted effect measures. Only five studies were identified which considered a population with underlying severe mental illness, with and without physical LTCs. In these studies, the addition of physical LTC increased the risk of hospital utilisation.

In populations with underlying diabetes, cardiovascular disease and COPD, people with SMI were at higher risk of 30-day readmissions compared to those without SMI, and the pooled OR were similar for 30-day readmission in these populations. This suggests that over this short timeframe, the risk of readmission does not differ substantially by underlying physical disease. While the effect size of having SMI was relatively small for all three diseases, any increased risk of hospital admission represents a major burden given the underlying high rate of admissions for these diseases in the general population [82, 83].

A strength of focusing on studies in populations with underlying physical LTC, is that it provides further evidence that the higher emergency hospitalisation in people with SMI is not due to higher prevalence of that LTC in the SMI population. It also allows the investigation of the impact of hospitalisations for the underlying LTC, compared to all-cause hospitalisations. In 30-day readmission studies of both COPD and cardiovascular disease we found little difference between studies of all-cause or cause-specific hospitalisations, suggesting that 30-day readmissions for the index condition are likely driving the difference between those with and without a diagnosis of SMI. The consistently higher risk in those with SMI, may indicate systematic differences in management and treatment of physical health conditions in people with SMI, such as lower adherence to medication, reduced access or attendance at planned outpatient care [84] and less guideline-recommended treatment [56, 61, 85–87], as well as more complex medication regimens and medical histories.

For studies examining hospital admissions for populations with underlying diabetes, we found that while patients with SMI had higher pooled OR of diabetes-specific admissions than those without SMI, the greatest difference was in all-cause admissions. This was also true in studies investigating both all-cause and diabetes admissions in the same study [41, 45]. These findings suggest that while a higher risk of diabetes admissions and sub-optimal management and treatment of diabetes [35, 37, 45] account for some of the higher hospital use in people with SMI, there are other factors involved. A study of patients with underlying diabetes found high rates of all-cause hospitalisations in people with SMI, even once acute psychiatric admissions were excluded from the outcome [45], suggesting that higher rates of multimorbidity, and therefore higher general physical health admissions, as well as higher risk of trauma and infectious disease hospitalisations [16], may be adding to the burden of hospitalisations in these patients. While we did not find the same in the subgroup analysis of diabetes studies presenting hazard ratios, only one study investigated all-cause admissions and the total number of available studies was small, limiting interpretation.

We also found evidence that specific populations may have elevated risk of hospital use. We found a high risk of hospitalisation in people with SMI in studies examining the effect of SMI on readmissions during hepatitis C treatment [65], on cardiovascular hospital use in African American patients with heart failure [59], on diabetes readmissions in patients under the age of 35 with type I diabetes [33], and following breast cancer surgery [64]. For diabetes, patients with schizophrenia appeared to be at higher risk of hospitalisation compared to other SMI diagnoses in studies presenting adjusted odds ratios. This has been reported elsewhere [17], and is in line with other studies that have found people with schizophrenia suffer more ill-

health, greater all-cause mortality and poorer physical health and treatment outcomes than people with other SMI diagnoses [9, 35, 37, 88, 89]. However, for studies of people with underlying diabetes or cardiovascular disease presenting adjusted hazard ratios, there was little difference between diagnoses of bipolar disorder and schizophrenia. Of the seven studies included in our review which considered schizophrenia alongside other SMI diagnoses, two found patients with schizophrenia were more likely to be hospitalised than other SMI diagnoses [31, 35], one found that those with schizophrenia were less likely to be hospitalised [36], and four found no significant difference [37, 59, 70, 71].

Finally, we found that while still elevated, the risk of readmission in patients with diabetes and SMI was lower in the US compared to other countries. While this finding has been documented before [17], the reason for this is unclear. For these studies, we found differences in effect size based on the healthcare system under investigation, and therefore patients with SMI may face different barriers and drivers to hospital use across payers in the US healthcare system. It is not clear whether this is limited to diabetes management, as the small number of studies in patients with COPD or cardiovascular disease did not permit comparisons by country.

## Limitations

Although this review has better described the pattern of hospital utilisation in people with SMI and physical health conditions, there are limitations. Although our search strategy was thorough, we may have missed studies which include SMI as a risk factor for higher healthcare utilization, but which do not include terms for SMI in the title or abstract. These studies are unlikely to have SMI as their main exposure variable and given that SMI is not common in the general population are less likely to provide well powered estimations. We identified 11 studies for which SMI was not the main focus, and while inclusion of these studies provides further evidence, caution is needed as they may be subject to confounding and issues of power [90]. In addition, while our search strategy was thorough, and overall agreement between reviewers was high (91%), the interrater reliability of screened abstracts as measured by the Kappa statistic was moderate (0.57). This is in part due to the large number of studies screened and the rarity of relevant studies [91], but also the complexity of multiple exposures and outcomes. All disagreements were discussed thoroughly to ensure the accuracy of study inclusion.

We found marked heterogeneity in the study results, particularly for studies of diabetes. While definitions of SMI, physical LTCs and outcome measures accounted for some of this, underlying differences in the population and healthcare system, as well as differences in study design are likely major causes of this heterogeneity.

While most studies we identified were of fair or good quality, there were limitations to many of them. Few studies utilised matched cohorts of patients, and most did not evaluate the impact of prior healthcare utilisation, despite this being a known predictor of hospital use in the general population [92]. Furthermore, many studies were performed in the US, which limits the generalisability of results to other healthcare systems. Despite being based in longitudinal populations, under half of studies performed a time-to-event analysis. Where this was performed, very few accounted for multiple hospitalisations or included time-varying covariates. Most studies included only patients who had accessed secondary care, both to define SMI and physical health conditions. Without access to primary care records, these studies exclude those patients who may be managed solely in primary care or attend secondary care very infrequently. These excluded patients may provide important information on protective factors that reduce secondary care use.

### Knowledge gaps and future research

There were few studies investigating hospital use in a population of patients with SMI, comparing hospital use in those with or without physical LTC. The underlying heterogeneity of these studies made them unsuitable for meta-analysis. Given that people with SMI are at an higher risk of many physical LTCs, further research is required to identify the drivers of physical health hospitalisations in people with SMI, and subsets of this population at higher risk.

There was also a lack of data regarding hospital use in patients with cancer, and the impact of SMI diagnoses on hospital utilisation. Given the higher risk of mortality following cancer diagnosis in those with SMI, and evidence of sub-optimal cancer screening and late diagnoses [93], it is important to understand hospital utilisation in this population.

Finally, there was a lack of information on the impact of SMI on hospitalisation for liver disease, and on the long-term risk of hospitalisation in patients with COPD or cardiovascular disease. These common diseases represent a huge burden in terms of hospital resource use and ill health in the general population [94]. Given people with SMI may be at higher risk of these diseases [2], receive poorer care [6, 56, 61, 84–87, 89, 95–98] and worse outcomes [6], more research is required into the impact of an SMI diagnosis on hospital utilisation in people with these conditions.

## Conclusions

This systematic review and meta-analysis found that patients with SMI and underlying physical health conditions are at a higher risk of hospital use for that condition, and for other causes. Further research is warranted into the effects of different physical health conditions and different SMI diagnoses on hospital use, particularly over longer time periods, and of pathways and drivers of hospitalisation in those with SMI. This will allow targeted interventions aimed at reducing inappropriate hospital use and improving disease management and outcomes in people with SMI.

## Supporting information

**S1 Checklist.**
(DOC)

**S1 Appendix. Search strategy.**
(DOCX)

**S1 Table. Study quality and detailed characteristics.** *Does control for age and is limited to females.
(DOCX)

**S2 Table. Components of the Newcastle-Ottawa score.** *One point. a: Analysis presenting odds ratios; b: analysis presenting hazard ratios c: Analysis of 30-day readmissions; d: analysis of long-term readmissions.
(DOCX)

**S1 Fig. Funnel plots for all individual analyses.**
(DOCX)

**S2 Fig. Funnel plot for studies presenting adjusted odds ratios of hospital utilisation in diabetes patients with SMI compared to without SMI.**
(DOCX)

**S3 Fig. Funnel plot for studies presenting adjusted hazard ratios of hospital utilisation in diabetes patients with SMI compared to without SMI.**
(DOCX)

**S4 Fig. Funnel plot for studies presenting adjusted odds ratios of hospital utilisation in heart disease patients with SMI compared to without SMI.**
(DOCX)

**S5 Fig. Funnel plot for studies presenting adjusted hazard ratios of hospital utilisation in heart disease patients with SMI compared to without SMI.**
(DOCX)

**S6 Fig. Funnel plot for studies presenting adjusted odds ratios of hospital utilisation in COPD patients with SMI compared to without SMI.**
(DOCX)

## Author Contributions

**Conceptualization:** Naomi Launders, Louise Marston, Gabriele Price, David P. J. Osborn, Joseph F. Hayes.

**Data curation:** Naomi Launders, Kate Dotsikas.

**Formal analysis:** Naomi Launders.

**Funding acquisition:** Gabriele Price, David P. J. Osborn.

**Investigation:** Naomi Launders, Kate Dotsikas.

**Methodology:** Naomi Launders, Louise Marston, Gabriele Price, David P. J. Osborn, Joseph F. Hayes.

**Supervision:** Louise Marston, Gabriele Price, David P. J. Osborn, Joseph F. Hayes.

**Validation:** Kate Dotsikas.

**Visualization:** Naomi Launders.

**Writing – original draft:** Naomi Launders.

**Writing – review & editing:** Naomi Launders, Kate Dotsikas, Louise Marston, Gabriele Price, David P. J. Osborn, Joseph F. Hayes.

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
