## [Decision Letter · Decision Letter 0]

14 Mar 2022

PONE-D-21-32052The impact of comorbid severe mental illness and common chronic physical health conditions on hospitalisation: A systematic review and meta-analysisPLOS ONE

Dear Dr. Launders,

Thank you for submitting your manuscript to PLOS ONE. After careful consideration, we feel that it has merit but does not fully meet PLOS ONE’s publication criteria as it currently stands. Therefore, we invite you to submit a revised version of the manuscript that addresses the points raised during the review process.

We look forward to receiving your revised manuscript.

Kind regards,

Michele Fornaro

Academic Editor

PLOS ONE

“This report is independent research supported by the National Institute for Health Research ARC North Thames. The views expressed in this publication are those of the author(s) and not necessarily those of the National Institute for Health Research, Public Health England, or the Department of Health and Social Care.”

“This study was supported by Public Health England (PhD2019/002 - NL), the Wellcome Trust (211085/Z/18/Z - JFH), the Medical Research Council (MC\\PC\\17216 - DPJO), University College London Hospitals NIHR Biomedical Research Centre (NL, DPJO, JFH) and the NIHR ARC North Thames Academy (DPJO, JFH).

This report is independent research supported by the National Institute for Health Research ARC North Thames.

Reviewers' comments:

Reviewer's Responses to Questions

**Comments to the Author**

1. Is the manuscript technically sound, and do the data support the conclusions?

Reviewer #1: Yes

Reviewer #2: Yes

2. Has the statistical analysis been performed appropriately and rigorously? 

Reviewer #1: Yes

Reviewer #2: Yes

3. Have the authors made all data underlying the findings in their manuscript fully available?

Reviewer #1: Yes

Reviewer #2: Yes

4. Is the manuscript presented in an intelligible fashion and written in standard English?

Reviewer #1: Yes

Reviewer #2: Yes

5. Review Comments to the Author

Reviewer #1: The present Systematic Review and Meta-analysis aimed at identifying the hospital utilization's rates in people affected by SMI with a comorbid physical health condition. The topic is interesting and the authors made a huge effort in summarizing the available evidence, choosing multiple outcomes and health conditions.

I have a few comments that, in my opinion, may help to improve or clarify the manuscript:

- The search was conducted until March 2020. Since 2 years have passed, I think that the authors should update their original search to detect the most current studies on the topic. Additionally, I suggest to put the complete data (dd/mm/yyyy) in the main text to allow an easier reproducibility;

- Which definition of SMI did the authors adopt? I think that they should specify how they define SMI. For example, they excluded the MDD from their SMI definition (line 89), while it is included as SMI in other studies;

- Did you include only those studies adopting DSM/ICD criteria to make a diagnosis of SMI? If not, why?

- Line 117, the authors used a score of 6 as a cut-off to divide low and high-risk of bias studies. I think that the authors should motivate their decision as it could seem arbitrary (i.e., citing other studies adopting the same method);

- The Cohen's Kappa is 0.57 that is quite low compared to other reviews. I think that the authors should double-check this passage and consider to add it as a limitation;

- Table 1, I think it could be useful to add a column in which reporting the study design;

- Line 308, could you please rephrase this sentence? It appears quite unclear to me

Reviewer #2: The authors appropriately performed statistical analysis. The results are exhaustively presented and adequately discussed in the light of the most recent literature evidence. Compared to the relevant meta-analysis performed by Ronaldson A. et al. (2020), the present MA has the advantage of assessing the outcomes separately for the five medical comorbidities examined, thus adding a relevant contribution to the field.

6. PLOS authors have the option to publish the peer review history of their article (what does this mean?). If published, this will include your full peer review and any attached files.

Reviewer #1: No

Reviewer #2: **Yes: **martina billeci

---

## [Author Response · Author response to Decision Letter 0]

5 May 2022

We thank the editors and reviewers for their comments. Detailed responses to reviewers are included in the submission and we have emailed the PLOS office regarding changing the funding options. A change to the funding statement is included in the cover letter and in the response to reviewers.

---

## [Decision Letter · Decision Letter 1]

21 Jul 2022

The impact of comorbid severe mental illness and common chronic physical health conditions on hospitalisation: A systematic review and meta-analysis

PONE-D-21-32052R1

Dear Dr. Launders,

We’re pleased to inform you that your manuscript has been judged scientifically suitable for publication and will be formally accepted for publication once it meets all outstanding technical requirements.

Kind regards,

Giuseppe Carrà, PhD

Academic Editor

PLOS ONE

Reviewers' comments:

Reviewer's Responses to Questions

**Comments to the Author**

1. If the authors have adequately addressed your comments raised in a previous round of review and you feel that this manuscript is now acceptable for publication, you may indicate that here to bypass the “Comments to the Author” section, enter your conflict of interest statement in the “Confidential to Editor” section, and submit your "Accept" recommendation.

Reviewer #1: All comments have been addressed

2. Is the manuscript technically sound, and do the data support the conclusions?

Reviewer #1: Yes

3. Has the statistical analysis been performed appropriately and rigorously? 

Reviewer #1: Yes

4. Have the authors made all data underlying the findings in their manuscript fully available?

Reviewer #1: Yes

5. Is the manuscript presented in an intelligible fashion and written in standard English?

Reviewer #1: Yes

6. Review Comments to the Author

Reviewer #1: The authors updated their search string and implemented most of my suggestions.

Thank you so much!

7. PLOS authors have the option to publish the peer review history of their article (what does this mean?). If published, this will include your full peer review and any attached files.

Reviewer #1: No

---

## [Editor Report · Acceptance letter]

9 Aug 2022

PONE-D-21-32052R1 

The impact of comorbid severe mental illness and common chronic physical health conditions on hospitalisation: A systematic review and meta-analysis 

Dear Dr. Launders:

I'm pleased to inform you that your manuscript has been deemed suitable for publication in PLOS ONE. Congratulations! Your manuscript is now with our production department. 

Kind regards, 

on behalf of

Dr. Giuseppe Carrà 

Academic Editor

PLOS ONE